# Assessment of Remediation of Municipal Wastewater Using Activated Carbon Produced from Sewage Sludge

**Khuthadzo Mudzanani [1,2], Sunny Iyuke [1] and Michael O. Daramola [3,***

[1] School of Chemical and Metallurgical Engineering, Faculty of Engineering and the Built Environment, University of the Witwatersrand, Johannesburg 2050, South Africa; sunny.iyuke@wits.ac.za (S.I.)

[2] Measurements and Control Division, Mintek, Praegville, Randburg 2194, South Africa

[3] Department of Chemical Engineering, Faculty of Engineering, Built Environment and Information Technology, University of Pretoria, Hatfield, Pretoria 0028, South Africa

\* Correspondence: michael.daramola@up.ac.za

**Abstract:** This study evaluates the potential to synthesize an adsorbent for wastewater remediation applications from an anaerobic digestion by-product synthesized using biomaterials and a less energy-intensive process. The synthesized sludge-based granular activated carbon (GAC) was used to adsorb Cr(VI) and Cd(II) in a batch reactor stirred for 24 h at 25 °C. The surface chemistry of the material was assessed porosity with BET, SEM for morphology, EDS-XRF for elemental analysis, and functional groups on these materials using FTIR and TGA for thermal profile. $S_{BET}$ of the SAC was discovered to be 481.370 $m^2/g$ with a $V_T$ of 0.337 $cm^3/g$, respectively 9.02 and 2.23 times greater than raw sludge. The modification to SAC shows a dramatic increase in performance from 40% to 98.9% equilibrium adsorption rate. The maximum or equilibrium removal (99.99%) of Cr(VI) and Cd(II) was achieved by 0.8 and 1.4 g SAC dosage, respectively. Thus, it can be concluded that activation of sewage sludge was effective in enhancing the surface area and pore volume which made it suitable for AMD remediation application.

**Keywords:** digestate; sludge-based activated carbon; adsorption; anaerobic digestion

## 1. Introduction

The increase in world population has resulted in increased industrialization and urbanization. Consequently, the volume of wastewater generated has increased annually, resulting in an increase in volumes of sewage sludge generated in the municipal wastewater treatment works (WWTW) [1]. Anaerobic digestion has been implemented as a solution for sewage sludge waste management. During the process, methane is produced and the digested material is generated as a by-product. Literature has reported that municipal sewage sludge contains elements that can be modified into a potentially good adsorbent for wastewater treatment [2–4].

On the other hand, wastewater is a concern to the health and environment of living organisms, including humans by imposing risk to human health and restricting access to clean water [5,6]. The most common and significant contaminants include organic matter, nutrients, pathogens, heavy metals, toxic chemicals, suspended solids, pH, oil and grease, pharmaceuticals and personal care products (PPCPs), and endocrine-disrupting compounds (EDCs) [7,8]. Amongst others, this study focuses on the removal of toxic heavy metals such as cadmium, and chromium that contaminate wastewater from industrial processes.

Hexavalent chromium (Cr(VI)) is a toxic and highly soluble form of chromium, causing environmental concerns in water sources [9]. Cadmium (Cd(II)) is also a toxic heavy metal that can cause severe health issues and environmental pollution [10,11]. Based on the municipality where this study was conducted, the most challenging heavy metals to remove are Cr(VI) and Cd(II). Common treatments include chemical reduction, precipitation, membrane filtration, ion exchange and adsorption. However, these methods

can be expensive, difficult to upscale and impractical for areas with limited resources. Disposal of waste generated from the treatment processes is a concern, and it is essential to find an alternative method that is environmentally friendly. Continuous monitoring and maintenance are necessary to ensure system effectiveness over time [12]. Further research is needed to develop innovative, cost-effective, sustainable treatment technologies for efficiently removing Cr(VI) and Cd(II) from water sources.

Adsorption is a widely used method for removing heavy metals from environmental matrices such as water and wastewater. It offers high removal efficiency, versatility, and wide availability of adsorbent materials. In addition, adsorption processes are simple to operate, require lower energy input, and generate minimal sludge [13,14]. Adsorbents can be regenerated and reused numerous times, making the process more environmentally friendly and cost-effective. This is more feasible especially when using locally available, natural or waste [3].

This has increased interest in studies on the synthesis of sludge-based activated carbon (SAC) material as an effective biomaterial for the adsorption of heavy metal ions or dyes from municipal, industrial, and acid mine drainage wastewater. Demirbas et al. [15], Nkutha et al. [16] and Garg et al. [17] studied the removal of Cr(VI) onto naturally derived adsorbents was showed high adsorption capacity at low temperatures. Similarly, Kour et al. [10], Abbou et al. [18] and Wang et al. [11] studied the removal of Cd(II) using biomaterial, natural clay and coal-based humin. However, at the time when this study was conducted, there was very limited literature reported that shows the removal of Cr(VI) and Cd(II) simultaneously using the municipal sewage sludge as a precursor to synthesize adsorption material.

Surface adsorption mechanisms involve metal and activated carbon interactions, where pollutants are absorbed into the material's pores. Activated carbon (GAC) is a renowned adsorbent for various contaminants in water treatment [19,20]. It is the most cost-effective and widely used material for removing organic micropollutants from wastewater treatment due to its large surface area, affinity for various organic compounds, and cost-effectiveness [20]. GAC's resilience, simplicity, environmental friendliness, low sludge production restrictions, and selectivity make it superior to other methods. As commercial pure GAC prices increase, wastewater treatment industries are seeking more cost-effective alternatives [14]. Thus, the implementation of SAC adsorption material in the removal of toxic heavy metals from wastewater is deemed a solution. This is due to the fact that it is synthesized from waste sludge, low-cost modification process, highly efficient, and has low installation, operations and maintenance costs [15].

This study looks at the feasibility of using SAC for the removal of heavy metals, Cr(VI) and Cd(II) in terms of its preparation or synthesis, characterization and performance evaluation. The specific aim of the present research is to investigate the adsorption characteristics of heavy metal ions by SAC material from sewage sludge. This addresses the valorization of municipal sewage sludge, which is available in abundance and solves the sludge management simultaneously. The research study reports on the effects of contact time and SAC dosage on the adsorption isotherm and the kinetics of adsorption of heavy metal ions are reported.

## 2. Materials and Methods

### 2.1. Pre-Experimental Characterization

The sludge samples were the digested materials from the settling tank of the anaerobic digester system from the nearby municipality, Johannesburg South Africa. The sample was dried in the oven and kept in the desiccator for at least 2 days until all the sample was moisture-free. The commercial GAC (CGAC) media was sourced from coal precursor materials, purchased from ChemLab, South Africa. These were subsequently used for characterization. The surface chemistry of activated carbon can be assessed by detecting specific functional groups using Fourier transform infrared spectroscopy (FTIR), XRF for elemental presence, SEM for morphology and BET for their surface area and porosity.

Activated carbon porosity was assessed based on $N_2$-gas adsorption and desorption at liquid nitrogen temperature in a BET instrument.

## 2.2. The Sludge-Based Activated Carbon Preparation

The sludge-based activated carbon was produced using modified methods adapted from Wang et al. (2019) [19]. To remove any dirt or inorganic contamination, the sludge sample was washed with deionized water and ethanol and dried at 60 °C in an oven until dry. The material is then crushed, mixed with clay for compounding, and passed through 1000–500 μm screens. A porcelain crucible containing 50 g of the material was heated in a tubular reheating furnace at 350 °C for 1 h under a nitrogen atmosphere. The prepared material was washed with KOH before being dosed with $ZnCl_2$ in a 4:1 mass ratio. 200 g sludge + 50 g $ZnCl_2$. The mixtures were then heated to 800 °C for 2 h.

Carbon materials were obtained by repeatedly washing them with distilled water. To regenerate the adsorbent, the material was centrifuged from the solution and calcined at 800 °C for 2 h in an $N_2$ atmosphere. The products were then collected by washing them repeatedly with very low concentrations of potassium hydroxide and distilled water. After drying the washed materials for 24 h, sludge-based activated carbon (SAC) was recovered. All the material was crushed and fractionated by sieving to obtain a 37.5–63.0 mm size fraction for the batch adsorption isotherm testing; this was carried out for all samples, including raw sludge, CGAC, and SAC material. After that, the media were thoroughly cleaned in ultrapure water, dried overnight at 105 °C, and stored in a desiccator until used in the adsorption capacity batch testing.

## 2.3. The Preparation of the Stock Solution

To prepare a stock solution of Cr(VI), 1000 mg of Cr(VI) salt (potassium dichromate) was mixed in 100 mL of distilled water in a volumetric flask to give a 10,000 ppm Cr(VI). For Cd(II), 1000 mg of Cd(II) salt was mixed with 90 mL of distilled water in separate 100 mL volumetric flasks to give a 10,000 ppm Cd(II) stock solution. To prepare 1000 ppm solutions, take 10 mL of each 10,000 ppm stock solution and dilute each with 90 mL of distilled water in separate 100 mL volumetric flasks. Mix thoroughly to obtain 100 mL of 1000 ppm solutions of Cr(VI) and Cd(II). Diluting the stock solution and adjusting the pH with HCl (0.1 M) and/or NaOH (0.1 M) solutions yielded the working solutions. The pH effect was assessed using 50.0 mL of ion solution with an initial concentration of 10 mg/L and a pH range of 2.0 to 8.0.

## 2.4. Batch Adsorption Study

In each batch adsorption experiment, a known quantity of the adsorbent is mixed with a specific volume and concentration of Cr(VI) and Cd(II)metal ion solution. The mixture was then agitated for 24 h at a constant temperature to allow adsorption to occur. During the experiments, several parameters were controlled, including pH (which was adjusted using acid/base solutions), temperature at 35 °C, adsorbent dosage, contact time, and initial metal ion concentration. The effect of each parameter on the adsorption efficiency was studied. After the adsorption process, the remaining concentration of Cr(VI) and Cd(II) in the solution was sampled and analyzed using an ultraviolet-visible spectrophotometer (UV-vis) instrument and when necessary, the HACH DR2800 (Loveland, CO, USA) water quality analyzer. The number of ions adsorbed per mass unit of the SAC (mg/g) was calculated using the calculation given in Equation (1) to determine the adsorption capacity:

$$Q_e = \frac{(C_i - C_e)}{m}V \tag{1}$$

where $Q_e$ represents the number of ions adsorbed at equilibrium (mg/g), $C_i$ represents the initial metal ions concentration (mg/L), $C_e$ represents the equilibrium metal ions concentration (mg/L), V represents the volume of the aqueous phase (L), m represents the amount of SAC material used (g), and $C_t$ represents the metal ions concentration at

time t (mg/L). Using NaOH and/or HCl solutions, the pH was changed to the appropriate value [4–12]. For adsorption kinetics, a 200 mL (5 mg/L) mixture of the test solution and SAC (20 mg) was continuously agitated in a shaker at 25 °C for 2 h. After 10, 15, 20, 30, 60, and 120 min, separate standard samples were obtained. The suspension was filtered before being subjected to UV-Vis analysis.

### 2.5. Characterizations

2.5.1. Thermogravimetric Analyses

The prepared materials' dehydration kinetics were determined using thermogravimetric analysis (TGA). The precursor's thermal decomposition was measured at a temperature ramp rate of 10 °C/min in an air environment between 25 °C and 1000 °C [19].

2.5.2. Scanning Electron Microscopy (SEM)

The morphology of the materials was examined using scanning electron microscopy (SEM), using a Philips XL30 SEM (Irvine, CA, USA) with a maximum voltage of 20 kV, to obtain preliminary images [21].

2.5.3. Fourier Transform Infrared Spectroscopy (FTIR)

The sample's functional groups were obtained with Fourier transform infrared spectroscopy (FTIR) and were acquired in the mid-infrared range (250–4000 $cm^{-1}$) using a Shimadzu 4800S (Burladingen, Germany). The spectrum was scanned with 20 scans at a resolution of 2.0 $cm^{-1}$ [22].

2.5.4. Energy Dispersion X-ray Fluorescence Spectrometer (ED-XRF)

The screening analysis of elements present in a raw sludge sample and SAC was performed using an Energy Dispersion X-ray fluorescence Spectrometer (ED-XRF) (Epsilon 3 XL, PANalytical) (New York, NY, USA). This device enables a semi-quantitative elemental examination of finely powdered substances for elements heavier than oxygen. For the analysis, a 10 mm thick sample was placed in a 28 mm diameter sample holder over a Mylar sheet (6.3 m) [23].

2.5.5. BET: Pore Size Distribution and Specific Surface Area

The nitrogen adsorption and desorption isotherm were used to calculate the Brunauer–Emmett–Teller (BET) surface area ($S_{BET}$), while an Autosorb-1 surface area and pore size analyzer were used to calculate the pore size distribution and specific surface area [24].

2.5.6. UV Visual Spectrophotometer

In UV–Vis, a beam with a wavelength ranging from 180 to 1100 nm is passed through a cuvette containing a sample of the solution. This UV or visible light is absorbed by the sample in the cuvette. The amount of light absorbed by the solution is determined by its concentration, the length of the light path through the cuvette, and the analyte's ability to absorb light at a certain wavelength. UV/Vis can quantify the concentration of a single analyte in a solution quantitatively [25].

## 3. Results

### 3.1. Characterization of Precursor, CGAC & SAC

3.1.1. The Textural Properties

The most significant attribute of any adsorbent is its adsorption capacity, which is directly proportional to its specific surface area and pore volume [26]. Table 1 shows the textural properties of raw sludge, SAC, and commercial granular activated carbon (CGAC). The textural parameters of the material were determined using $N_2$ adsorption/desorption isotherms and the Brunauer–Emmett-Teller equation (BET). The total pore volume ($V_T$) of the dry medium was derived using BET theory as the adsorbed volume of $N_2$ gas approaching the saturation volume ($P/P_0 = 0.98$) and the surface area, $S_{BET}$.

**Table 1.** Characteristics of the porous structure of the raw sludge and SAC.

| Sample | $S_{BET}$ (m²/g) | $V_T$ (cm³/g) | $V_{MIC}$ (cm³/g) | $V_{MES}$ (cm³/g) | $D_P$ (nm) |
|---|---|---|---|---|---|
| Raw sludge | 53.389 | 0.148 | 0.019 | 0.010 | 7.627 |
| SAC | 481.370 | 0.337 | 0.341 | 0.21 | 4.814 |
| CGAC | 986.601 | 0.513 | 0.155 | 0.120 | 16.1 |

Density functional theory (DFT) was used to calculate the DP distribution for pores with diameters ranging from 0.7 to 36 nm. For non-homogeneous liquids over microporous surfaces, the DFT model was used to offer a more precise interpretation of the data. The findings in Table 1 reveal fast $N_2$ adsorption at low relative pressures, which is a feature of a microporous material. The capillary condensation that happens in the mesopores is connected with the adsorption hysteresis loop that appears in the medium relative pressures [27,28].

The surface feature for the CGAC adsorbent $S_{BET}$ has been previously reported as being 841 m²/g and $V_T$ 0.430 cm³/g, respectively [3,4,29]. In the current study, the CGAC was shown to be higher in strength with many pores, 986 m²/g and $V_T$ 0.513 cm³/g respectively. This is more than double the $S_{BET}$ of SAC with a value of 481.37 m²/g with a $V_T$ of 0.337 cm³/g and 53 m²/g of raw sewage sludge. The $D_P$ of SAC was 4.814 nm, which was much lower than that of the raw sludge (7.627 nm). The $S_{BET}$ of the CGAC & SAC was found to be 986 m²/g and 481.370 m²/g, both of which were almost 18.6 and 9.2 times higher than those of the raw sludge, respectively.

This demonstrates that activating any material increases the surface area and porosity of the material by transforming mesopores or macropores into micropores [28]. As previously reported, by the continuous increase in temperature to activate the material results in the breaking down of the hemicellulose/cellulose and conversion into pyrolytic gas which results in the formation of pores [30]. Thus, the increase in the surface area, $S_{BET}$ and pore volume of the material of SAC was due to the activation that improves the carbon content in sludge [31]. Similarly, to SAC the CGAC material is coal heated at high temperature, above 400 °C to alter the surface structure mainly by adding pores [11]. This is the most likely reason for the significant increase in the surface area and pore volume of the material.

As outlined in the methodology, the raw materials were washed with KOH solution and were sufficiently dosed with a low concentration of $ZnCl_2$. The impregnation of SAC material with $ZnCl_2$ results in Cl resting on the outer layer or the surface of the material. This causes a localized breakdown, which extends the pore size. Due to this process, mesopore production is enhanced, and the resultant SACs have relatively large surface areas. When compared to ordinary activated carbons, the SACs generated had a comparatively large micropore volume ($V_{MIC}$) [32]. The addition of $ZnCl_2$ is involved in two conflicting mechanisms: micropore formation and pore widening [33].

### 3.1.2. SEM: Morphology Analysis

The morphology of the materials was evaluated using SEM. According to the approach outlined earlier in the experimental section, micrographs were obtained from CGAC, raw sludge, and carbonized sewage sludge, SAC. These micrographs were chosen to provide a typical representation of the observed structures; the magnification may vary to emphasize certain regions of interest. As shown in Figure 1A, the raw sewage sludge has a compact structure with no cavities. After activation, the SAC shown in Figure 1B has a loose structure in which the surface of the carbon is decorated with macropores. The CGAC surface structure looks compact, it is similar to that of a raw sludge, see Figure 1C. However, when it has been zoomed closer at 1000× magnification, the structure in Figure 1D is more porous and appears to have spherical uniformity that can serve as adsorption sites for adsorbates.

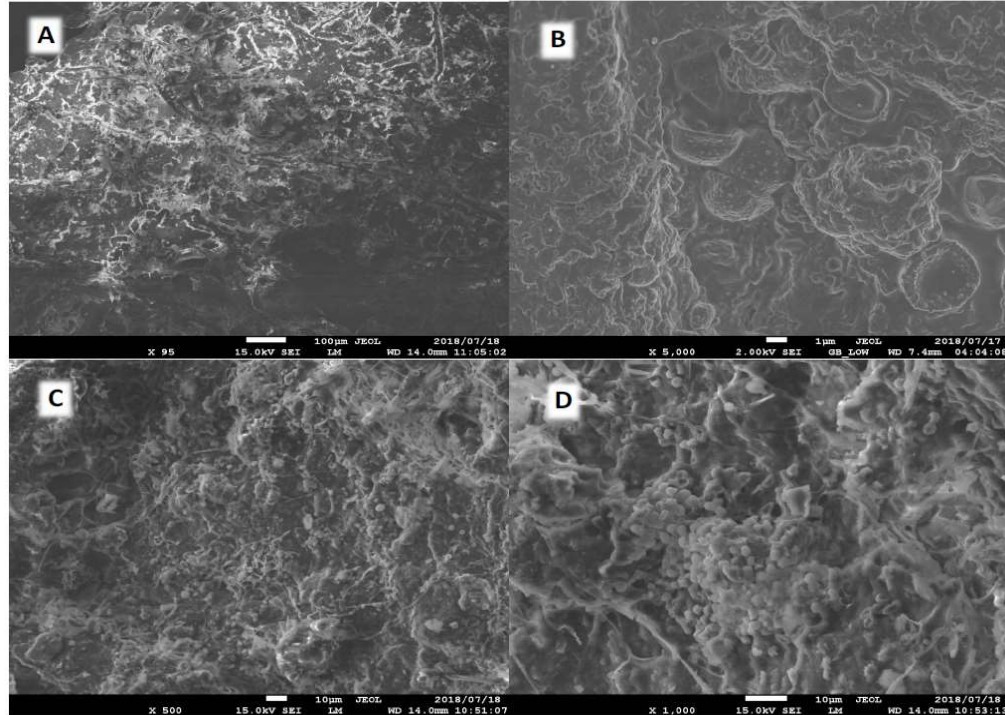

**Figure 1.** SEM micrographs, (**A**): Raw sludge (**B**): SAC shows spherical carbon particles on the surface of carbonized (**C,D**): CGAC.

As can be seen in Figure 1, the raw sludge material (A) does not differ structurally from the SAC itself (B). However, on the surface of the particles even at a magnification of 5000×, carbon spheres are visible (Figure 1B). The polymer's regular spherical form is well-kept after activation, which can be attributed to the chloromethylated spherical polymer's additional post-crosslinking during activation. This boosts the sphere's strength and thermos stability. The micrograph (×5000) reveals a dense network of interspaces on the scale7 of nanometers, indicating a highly developed disordered porosity [34].

These findings were consistent with the textural features shown in Table 1, which demonstrated that the surface area increased after modification owing to the activation and release of volatile chemicals during the carbonization process. BET results show the attack in the surface and the lignocellulosic structure of the raw sewage sludge resulting in the more porous structure of SAC. Figure 1A presents a dense surface whereas SAC and CGAC in Figure 1B,D depict a rougher surface with voids of varying sizes, suggesting a well-developed porous structure. The pores on the surface of SAC provide ideal routes for adsorbate ions to adhere into the carbon structure, enabling access to its mesopores and micropores where they can interact with the surface functional groups [19,35].

### 3.1.3. TGA: Thermogravimetric Analysis

The thermogravimetric analysis of the sewage sludge precursor, CGAC and SAC represented in Figure 2 shows the mass loss percentage versus temperature. The amount of weight loss or gain observed during TGA experiments can be correlated with the amount of organic matter present in each material, a link to adsorption capacity and adsorption mechanisms. In addition, the TGA data can complement the isotherm and kinetic studies as well as validate results obtained from SEM-EDS and BET surface analysis [36,37]. The decomposition kinetics of the organic matter in the precursor raw sludge was faster than CGAC and SAC. As shown in Figure 2, the weight loss in phase 1, range of 50 °C–200 °C corresponds to the dehydration of the materials meaning the elimination of both surface and lattice water [21].

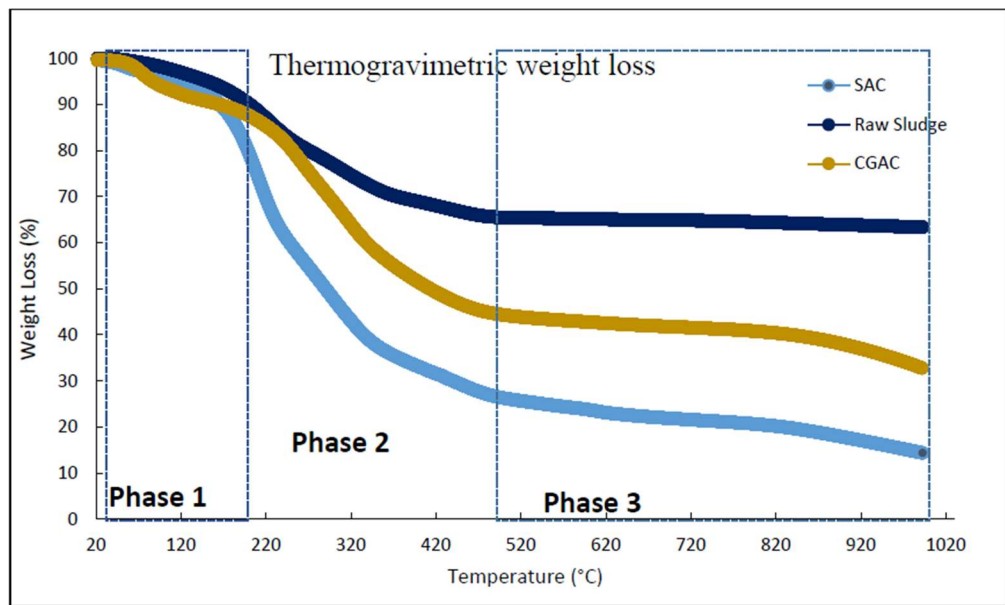

**Figure 2.** Thermal decomposition of raw sewage sludge and SAC.

The pyrolytic reactions take place mainly from 200 °C to about 500 °C, this is when the material is converted from the organics to the pyrolytic gas which results in pores, as mentioned previously [38]. This integrates with the BET surface area analysis, which exhibited increased porosity from the precursor (raw sludge) to SAC to CGAC material. As previously stated, the material with a high surface area has the potential to be an excellent adsorbent material due to an abundance of accessible sites to adsorb the heavy metal ion [39].

The weight loss in phase 2, between 200 and 550 °C can be attributed to the elimination of different vapors such as $NH_3$, $NO_2$ and $O_2$ [40]. This phase also shows high decomposition kinetics of the organic matter in the precursor raw sludge was faster than CGAC and SAC. The mass loss was lower but non-negligible in the range between 500 °C and 1000 °C. The plateau at higher temperatures (phase 3, above 550 °C) corresponds to the existence of some oxide minerals in the samples ($SiO_2$, MgO and $Al_2O_3$) which do not break down or decompose [37]. According to the research conducted by Kristl et al. [41] mass loss over 600 °C might be attributable to carbonate thermal degradation.

The TGA gave an indication of the temperature at which the adsorbent begins to degrade or lose its structural integrity. In this case, it is represented by phase 3 in Figure 2, and the synthesized material SAC degrades easily at temperatures exceeding 500 °C. This indicates an incapacity to be regenerated by the calcination process, which is typically performed at temperatures exceeding 800 °C [22]. This is considered crucial, especially when researching the treatment of wastewater with adsorbents. This information aids in understanding the material's limits and prospective usage at high temperatures, where it may be exposed to various effluent streams [42]). In summary, TGA results show weight losses of ~30.64%, ~64%, and 80% were observed for raw sludge, CGAC and SAC material, respectively. This was due to the integration of moisture weight, organic matter, oxides and carbonate thermal degradation, the remaining ash was inorganic matter.

### 3.1.4. FT-IR Analysis

To analyze and compare the chemical bonding in the CGAC, raw sludge, and SAC material, Fourier transform infrared (FT-IR) spectra were obtained. Figure 3a–c shows the FTIR spectra of the CGAC raw sludge and SAC samples, respectively. Figure 3a shows that CGAC exhibits four primary absorption bands in the wavelength range of 4000–400 cm$^{-1}$. The O-H stretching vibrations cause the absorption maxima at 3400 cm$^{-1}$ and 1600 cm$^{-1}$.

The absorption peak at 1726 cm$^{-1}$ is caused by skeletal stretching vibrations, whereas the absorption peak at 1060 cm$^{-1}$ is caused by C=O stretching vibrations [43].

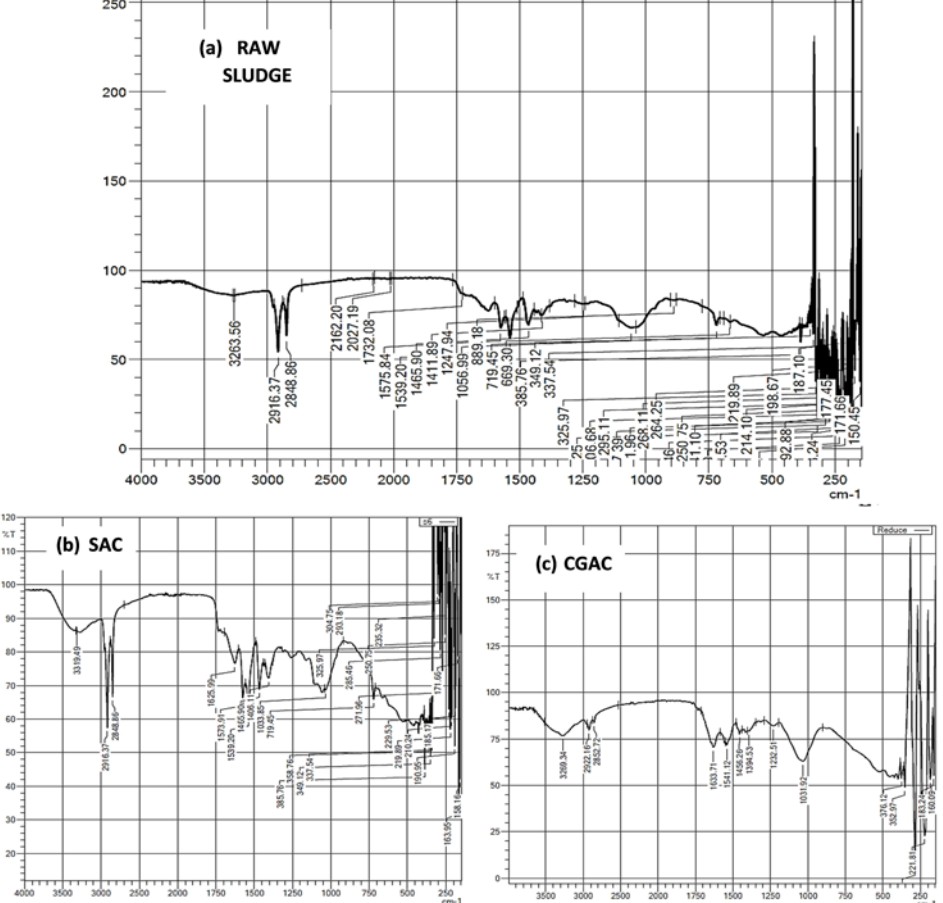

**Figure 3.** FTIR spectra of (**a**) raw sewage (**b**) SAC and (**c**) CGAC samples.

Figure 3b shows that raw sludge has limited stability, as evidenced by noise below 600 cm$^{-1}$, whereas Figure 3c shows that SAC material exhibits the characteristic of C=O and C-O stretching vibrations at 1650 cm$^{-1}$ and 1425 cm$^{-1}$, respectively. However, the additional peaks at 966 cm$^{-1}$ confirm the development of an N-H bond, indicating that nitrogen was successfully doped in the carbon framework. Meanwhile, the rise at 1020–1360 cm$^{-1}$ can be explained by the stretching and bending vibrations of nitrogen-containing functional groups such C=N and C-N and bending vibration of nitrogen-containing functional groups [35,44].

It is widely known that functional groups on the surface of AC, such as carboxyl, carbonyl, phenolic hydroxyl, and lactones, have a considerable influence on its adsorption ability [45]. The O-H stretching vibrations, which correspond to related peaks of hydroxyl groups from phenols and alcohols, are responsible for the wide absorption band in the 3300–3600 cm$^{-1}$ range [45]. The characteristic peak at 1624 cm$^{-1}$ can be ascribed to carbonyl's C=O and C=C vibration in carboxylic or ester groups [46]. The lactone groups appear to be responsible for the absorption peak at 1383 cm$^{-1}$. Alcohol groups are responsible for the rather strong band at 1042 cm$^{-1}$ (C-OH). In addition, the peaks at 665–200 cm$^{-1}$ represent the C-H stretching vibration. In this study, the activation led to an increase of oxygen-containing functional groups (C-OH, O-H and C=O), which might be attributed to the use of KOH and ZnCl$_2$ during synthesis [47].

### 3.1.5. ED-XRF Analysis

In elemental form, sewage sludge is a mixture of nitrogen- and Sulphur-containing organic materials, microorganisms and minerals, mainly C, O, Mg, Si, Ca, and Cl. It also contains trace elements such as Ti, Fe, Ba, Zn, and Cr. Figure 4 depicts an element screening analysis of GAC, dry raw sludge sample, and SAC using an energy dispersion X-ray fluorescence spectrometer (ED-XRF) (Epsilon 3 XL, PANalytical). Semi-quantitative elemental analyses of finely ground solids for elements heavier than oxygen are possible with this equipment [48]. Sludge contains fertilizing nutrients such as phosphorus, potassium, magnesium, sodium, calcium, and sulfur, as expected. The high potassium and chlorine concentration results from the use of zinc chloride for dosing during synthesis and washing of the SAC material with KOH. The SAC material was washed numerous times, hence the increase in low concentrations of most elements as compared with raw sewage sludge [23,47–49]. The presence of carbon was due to the heating activation that results in carbonization, similarly to the CGAC. The material was centrifuged from solution and calcined for 2 h in a $N_2$ atmosphere hence the presence of trace N in the SAC material.

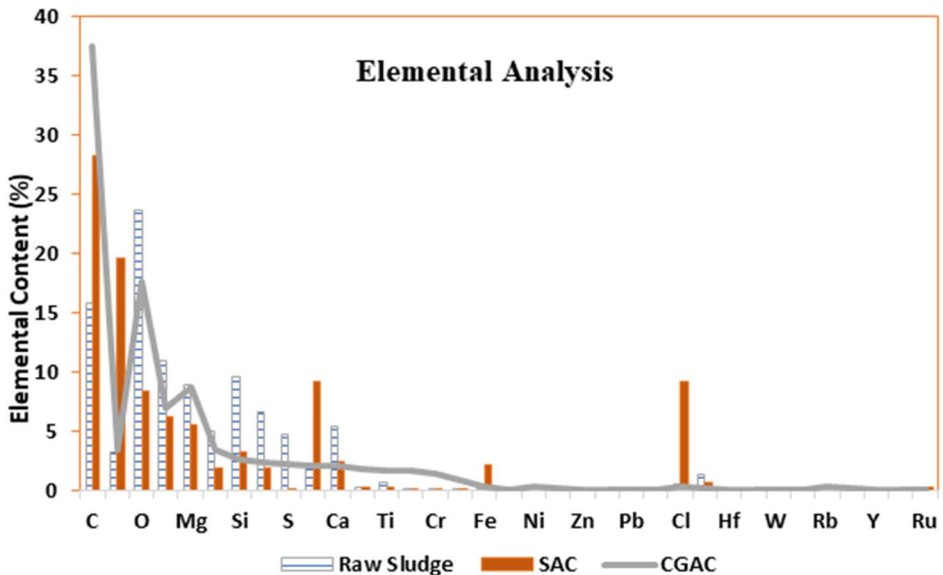

**Figure 4.** EDX spectra or Semi-quantitative XRF analysis of the raw sludge SAC and raw sludge samples.

### 3.2. Material Performance: Comparison

Comparison of SAC against CGAC and Raw Sludge

According to the literature, an assessment of the mechanisms of activated carbon adsorption can be performed in batch shake-flask experiments using an equilibrium approach, from which adsorption isotherms can be developed in order to obtain the material's maximum uptake capacity. In order to investigate the performance of sludge-based activated carbon for adsorption of hexavalent chromium (Cr(VI)) ions from aqueous solution, Cr(VI) solution was used. To compare the performance of SAC material against what is already in the market, commercial GAC (CGAC) was used. Furthermore, to see the effect of modifying the raw sludge into SAC, dried municipal sewage was tested under same conditions. Figure 5 shows the adsorption removal efficiency of Cr(VI) when using SAC, CGAC and raw sludge as the adsorption materials in different adsorption times ranging from 0 to 24 h.

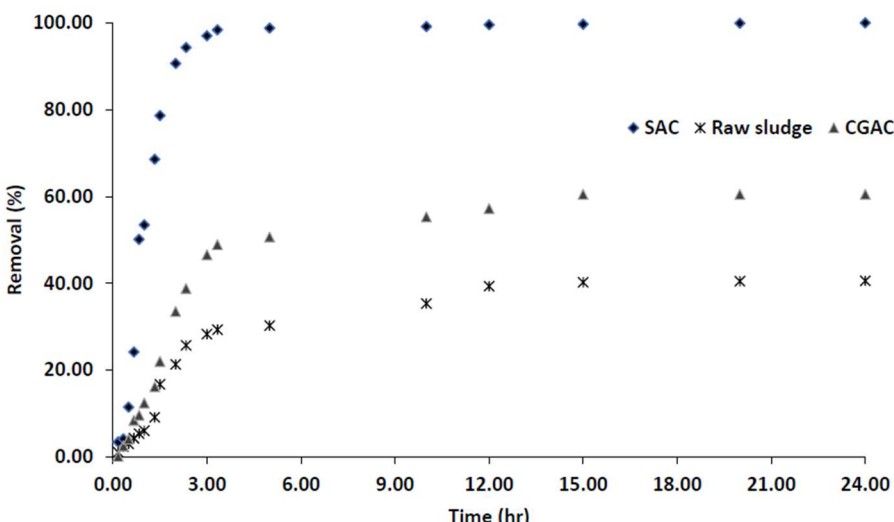

**Figure 5.** Adsorption of Cr(VI) by the sewage sludge-based activated carbon with time.

As it can be observed in Figure 5, the adsorption time has a significant impact on Cr(VI) adsorption, the results are comparable with what was describes in literature [13,50]. In batch tests, the concentration in solution of each of the compounds considered decreased continuously over time until an equilibrium point was reached for all adsorbents. This equilibrium point is a property of both the material and the adsorbate. The results show that when comparing SAC and raw sludge, the SAC achieved the highest Cr(VI) removal efficiency of 99.99%, while the raw sludge achieved only 40.71% removal efficiency. The increased surface area of the sludge material as reported in Section 3.1.1 corresponds with an increase in SAC removal efficiency. From these findings, the SAC can be regarded as a highly effective adsorbent as compared to raw sludge.

As indicated in Figure 5, the removal efficiency of Cr(VI) with CGAC and SAC went up to adsorption equilibria of 99.98%, and 60.50%, respectively. The SAC adsorbent reached equilibrium sooner than raw sludge and also exhibited a higher removal capacity for Cr(VI) followed by CGAC which is due to the availability of the adsorption sites in the material leading to the longer equilibrium time as observed in Figure 5. As a result, these compound molecules took longer to initiate adsorption, but once started, they achieved a higher level of adsorption [13,42].

*3.3. SAC Performance: Effect of the Operation Conditions*

3.3.1. Effect of the Initial pH

Figure 6 depicts the results of an investigation into the sorption efficiency of adsorbate onto SAC material as a function of pH (1–11). The pH of the solution has a significant impact on the adsorption of metal ions and dyes. This is due to the fact that the adsorbent's surface charge density and metallic speciation are pH-dependent. The results show that the removal efficiency of Cr(VI) decreases as pH increases as reported elsewhere [17]. At low pH (pH 1–3), a removal efficiency of 99.99% was achieved with the SAC whereas only 1% removal efficiency was achieved for Cd(II). However, at higher pH (e.g., at pH of 10) high adsorption was achieved.

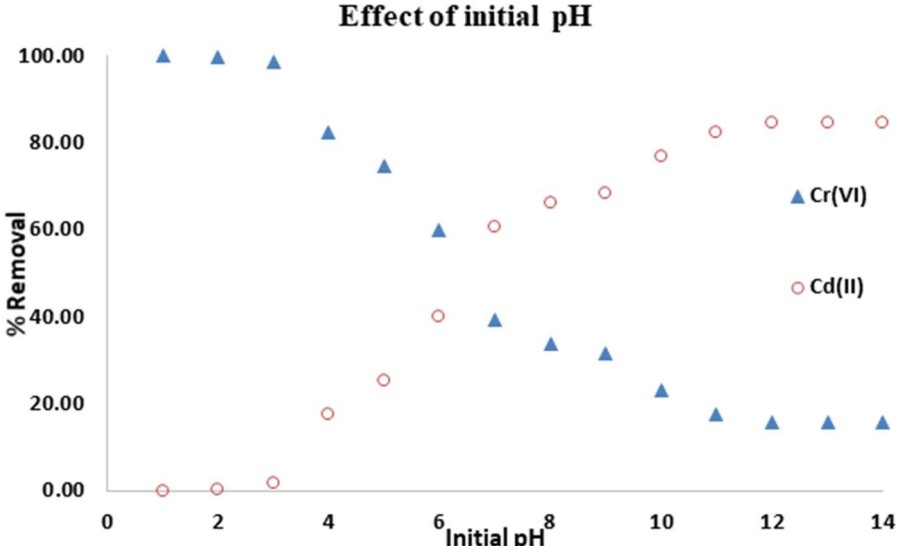

**Figure 6.** Effect of initial pH on adsorption of Cr(VI) and Cd(II).

It became necessary to investigate the electrostatic attraction/repulsion, chemical interaction, and ion exchange within the system to explain the observed adsorption behavior with varying pH. The adsorption capacity of Cd(II) increases with an increase in the initial pH in the solution, regardless of adsorbent type, according to the literature [10]. Due to the high concentration of hydronium ions ($H^+$) on the material surface at low pH, the affinity for positively charged $Cd^{2+}$ anions decreases since they compete for the same adsorption sites [18,51]. Conversely, at higher pH values, the presence of $H^+$ ions is low. Consequently, a greater number of ligands with negative charges (i.e., $OH^-$) will result in greater Cd(II) adsorption [18,52].

According to Cr(VI) speciation, the monovalent bichromate ($HCrO_4$) and divalent dichromate ($Cr_2O_7^2$) ions are the dominant species of Cr(VI) in aqueous solutions in the pH range 2–6, while the chromate ($CrO_4^2$) ion is the dominant species at pH > 6 [17,20]. Due to the high concentration of H+ on the material surface at low pH, the material surface becomes positively charged, increasing the affinity for the negatively charged $HCrO_4$ and $Cr_2O_7^2$ anions to bind on the clay without competing for adsorption sites [16,51]. More OH is present in the solution as the pH rises. This competes for adsorbent active sites with the chromate ions ($CrO_4^2$), resulting in low Cr(VI) removal [15].

### 3.3.2. Effect of SAC Dosage

The effect of SAC dosage on the removal of Cr(VI) and Cd(II) from aqueous solutions was studied. Figure 7 depicts the information. This is a critical parameter since it determines an adsorbent's capacity for a given initial concentration. In a large-scale industrial application of adsorbent in the removal of a desired solute, the quantity or mass of adsorbent is an important factor. A series of adsorption experiments were carried out with different masses of adsorbent at a fixed initial solute concentration to investigate the effect of adsorbent mass on adsorption [20,53].

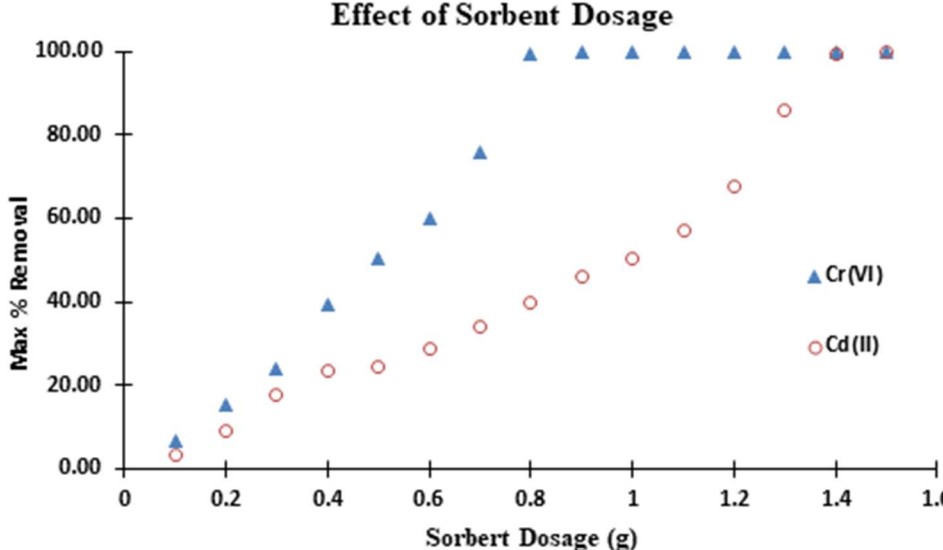

**Figure 7.** Effect of adsorbent dosage on adsorption of Cr(VI), Cd (II. Conditions: pH = 10 for Cd(II); pH = 2.0 for Cr(VI); 250 mg/L adsorbate concentration; 24 h. contact time.

In both cases, the results show that as the mass of the adsorbent increases, solute removal efficiency increases. The contact surface of adsorbent particles increases as the mass of the adsorbent increases, implying that more solute molecules will be adsorbed on the adsorption sites, resulting in an increase in adsorption efficiency. The removal, on the other hand, increases until it reaches a certain point, at which point it becomes constant. This limit refers to the maximum amount of ion that an adsorbate can absorb. The SAC dosage equilibrium removal can be concluded at this point.

The adsorption results show that as the SAC dosage is increased from 0.1 to 1.0 g, the percentage removal of Cr(VI) and Cd(II) increases gradually. At 0.1 g of SAC, exactly 10.8% Cr(VI) and Cd(II) were detected. With 0.8 and 1.4 g SAC dosages, maximum or equilibrium removal (99.99%) of Cr(VI) and Cd(II) was achieved, respectively. This is due to the fact that as the sorbent dosage is increased, there are more active adsorption sites available for the adsorbate to adhere to, resulting in a high percentage of removal [54]. The results show that a minimum of 0.8 and 1.4 g SAC dosage is required for quantitative removal (99.99%) of respective adsorbate with 250 mg/L concentration in 250 mL solution. As such, this adsorbent dosage was chosen for all subsequent batch experiments where applicable.

### 3.3.3. Effect of Initial Concentration on Adsorption Equilibrium

Figure 8 shows the adsorption uptake versus the adsorption time at various initial Cr(VI), Cd(II) concentrations at 30 °C. It indicated that the contact time needed for the solutions with initial concentrations of 50 mg/L to reach equilibrium was around 1–2 h. Such a rapid uptake is indicative of readily available sorption sites in the SAC material. Adsorption normally occurs in three steps and this behavior was observed for all the explored initial concentrations. At an initial concentration of 50 mg/L for instance, the first step (0–1 h 25 min) was very fast and characterized by the rapid attachment of adsorbates onto the surface of the SAC material [52,55].

The solutions with initial concentrations of 250 mg/L, required 2.5–3.0 h to reach equilibrium. The adsorption was slowed down as the reaction approached equilibrium. It is noted that this step (85–180 min) was slower due to intra-particle diffusion of the adsorbate to the internal matrix of the adsorbent [51]. More than 5 h was required for 500 mg/L initial concentration solutions to reach equilibrium.

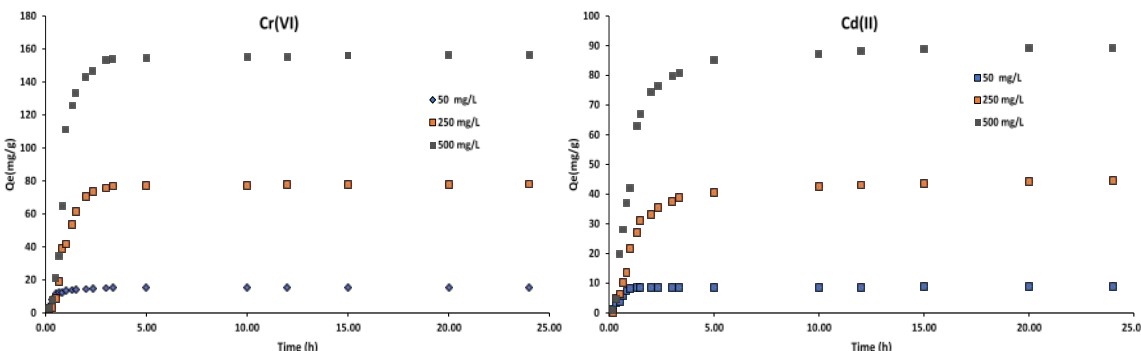

**Figure 8.** The adsorption uptake versus adsorption time at various initial concentrations at 30 °C on sludge-based activated carbon.

It takes longer to attain equilibrium at higher concentration since molecules have to first encounter the boundary layer effect and then diffuse from the boundary layer film onto the adsorbent surface [56]. Then finally, they have to diffuse into the porous structure of the adsorbent. Hence, the solutions of higher initial concentrations will take relatively longer contact time to attain equilibrium due to higher amount of heavy metal molecules [57,58].

The amount of dye or ion adsorbed on the SAC increases over time until it reaches a constant value beyond which dye removal from the solution is no longer possible [52,59]. The adsorption reaction will then come to a halt, indicating that the system has reached equilibrium. The maximum adsorption capacity of the adsorbent under those operating conditions is represented by the amount of dye adsorbed at the equilibrium time. With an increase in initial dye concentrations from 50 to 500 mg/L, the adsorption capacity at equilibrium ($Q_e$) was found to increase from 8.91 to 312.50 mg/g in this study. The mass transfer driving force would increase as the initial concentration increased, resulting in higher adsorption [55].

### 3.3.4. Adsorption Isotherm: Effect of Temperature

The adsorption isotherm is useful for describing how solutes interact with adsorbents and for optimizing adsorbent use. The fitting of adsorption equilibrium data into various isotherm models is also an important step in determining a model that can be used for design purposes [32,58]. The time series plots and kinetic curves of the adsorption process were also used to develop kinetic parameters. The Langmuir, Freundlich, Temkin, and Dubinin-Radushkevich isotherm models were used to study adsorption isotherms. The isotherm equation's applicability to describe the adsorption process was assessed by the correlation coefficients, $R^2$ values.

### 3.3.5. Langmuir Isotherm

For the Langmuir isotherm, when $C_e/Q_e$ is plotted against $C_e$, a straight line with slope of $1/Q_o$ and intercept of $(1/Q_o) K_L$ is obtained. The linearized Langmuir isotherm which is based on a monolayer sorption on the adsorbent surface with identical sorption sites is represented by Equation (3). The date is plotted in Figure 9 and the summary is shown in Table 2. Langmuir isotherms Equation (2):

$$\frac{C_e}{Q_e} = \frac{1}{Q_O}C_e + \frac{1}{Q_O K_L} \tag{2}$$

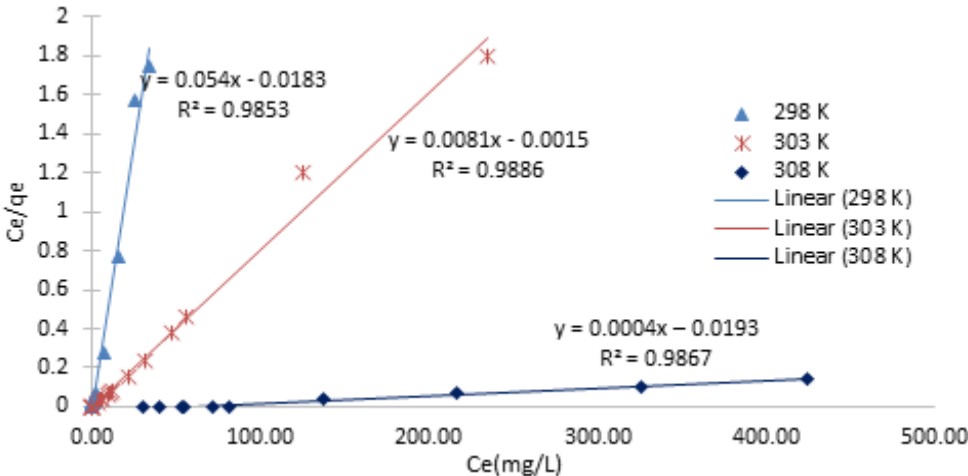

**Figure 9.** The Langmuir linear plots of $C_e/Q_e$ versus $C_e$.

**Table 2.** Summary of Langmuir Isotherm model constants and $R^2$ values obtained from the linear plot.

| Solution T (K) | Equation | $R^2$ | $1/Q_o$ | $Q_o$ | $1/Q_oK_L$ | $K_L$ |
|---|---|---|---|---|---|---|
| 298 | y = 0.054x − 0.0183 | 0.985 | 0.054 | 18.519 | 0.018 | 0.339 |
| 303 | y = 0.081x − 0.0015 | 0.989 | 0.081 | 12.346 | 0.002 | 0.019 |
| 308 | y = 0.0004x − 0.0193 | 0.987 | 0.0004 | 2500.00 | 0.019 | 48.250 |

The maximum number of ions per unit mass of adsorbent is presented by $Q_o$ (mg/g), this are ions required to form a complete monolayer on the adsorbent surface. $K_L$ is the rate constant related to the adsorption capacity and intensity of adsorption.

### 3.3.6. Freundlich Isotherm

The plot of log $Q_e$ versus log $C_e$ for the Freundlich isotherm yields a straight line with a slope of $1/n$ and an intercept of log $K_F$. The Freundlich isotherm is an empirical model based on adsorption on a heterogeneous surface, with an equation expressing the linear form in Equation (3).

Freundlich isotherms Equation:

$$\log(Q_e) = \log(K_F) + \frac{1}{n}\log(C_e) \tag{3}$$

where $K_F$ (mg/g) and $1/n$ constants are related to the adsorption capacity and intensity of adsorption.

The adsorption capacity and intensity of adsorption are related by the $K_F$ (mg/g) and $1/n$ constants. The isotherm models used, as well as all the constants and $R^2$ values obtained from each plot, are shown in Tables 2 and 3 and Figures 9 and 10. In comparison to the Freundlich model, the Langmuir model provided the best fit, with an $R^2$ value of 0.98 or higher.

**Table 3.** Summary of Freundlich Isotherm model constants and $R^2$ values obtained from the linear plot.

| Solution T (K) | Equation | $R^2$ | $1/n$ | $\log K_F$ | $K_F$ |
|---|---|---|---|---|---|
| 298 | y = −0.0506x + 1.4786 | 0.982 | 0.051 | 1.479 | 30.102 |
| 303 | y = −0.008x + 2.1885 | 0.717 | 0.008 | 2.189 | 154.348 |
| 308 | y = −0.0112x + 2.4865 | 0.821 | 0.011 | 2.487 | 306.549 |

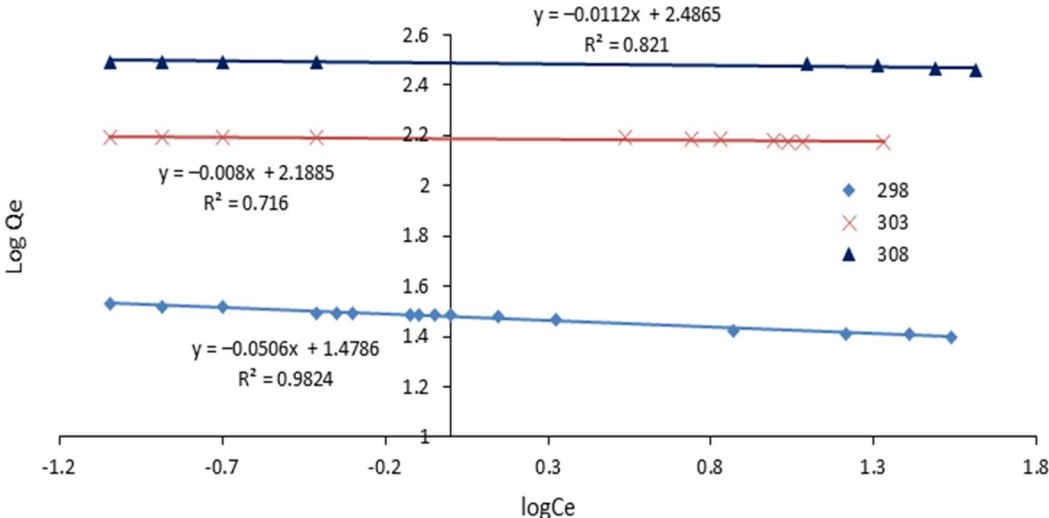

**Figure 10.** Linearized Freundlich isotherm for adsorption.

The homogeneous nature of the SAC surface was revealed by fitting the experimental data into the Langmuir isotherm equation. The results also showed the formation of a monolayer of dye molecules on the activated carbon's outer surface. Similar findings have been made elsewhere [39,60]. The adsorption capacity of the activated carbon used in this study was relatively high (312.45 mg/g).

## 4. Conclusions

The study carried out was to evaluate the feasibility of synthesizing and developing a sludge-based activated carbon, SAC from municipal sewage sludge and test its performance on the removal of Cr(VI) & Cd(II) a heavy metal, from wastewater. The synthesis technique to modify waste sludge into SAC was developed to produce activated carbon from sewage sludge. The main physicochemical findings were:

- The physical activation and the use of $ZnCl_2$ as the chemical activation were successful in increasing the surface area and enhancement of the porous structure as indicated by the SEM-EDS, TGA, FTIR and EDX results.
- The test performance carried out revealed that batch adsorption studies reported that the highest Cr(VI) removal efficiency was achieved with the SAC by 99.99%, whereas 60.50% and 40.71% removal efficiency was achieved from CGAC and raw sludge, respectively.
- The study on the effect of pH revealed that the competition between H+ and metal ions at low pH values are the main leading factors that affect the adsorption characteristics of SAC.
- The maximum or equilibrium removal (99.99%) of Cr(VI) and Cd(II) was achieved by 0.8 and 1.4 g SAC dosage, respectively.

**Author Contributions:** The manuscript was developed and conceptualized by K.M. and M.O.D. The supervision, review and correction of the manuscript was done by S.I. and M.O.D. All authors have read and agreed to the published version of the manuscript.

**Funding:** This research was funded by the Council for Scientific and Industrial Research, CSIR, South Africa under the studentship programme.

**Institutional Review Board Statement:** Not applicable.

**Informed Consent Statement:** Not applicable.

**Data Availability Statement:** Data will be made available on request.

**Acknowledgments:** I would like to acknowledge the University of Johannesburg, PEETS Doornfontein and the School of Chemical and Metallurgical Engineering, University of the Witwatersrand for allowing me to use their facility. My acknowledgment also goes to R. Idem, University of Regina Canada for showing interest in the work and providing directions.

**Conflicts of Interest:** The authors declare no conflict of interest.

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
