# Peer review of "Assessment of Remediation of Municipal Wastewater Using Activated Carbon Produced from Sewage Sludge"

_fermentation, doi:10.3390/fermentation9080769_

Round 1

Reviewer 1 Report

This study evaluates the potential to synthesize an adsorbent for wastewater remediation 10 application, from anaerobic digestion byproduct via a non-toxic and less energy intensive process. The topic is interesting and the result was well described. Minor revision is suggested. some minor suggestions is shown below:

Q1. delete the repeated word (from).

Q2. the legend in figure 5 is missing. 

Q3. would the Al element can be detected in these sludges (Figure 4)?

Author Response

The attached document contains the responses to the comments from the reviewer 1.

Reviewer 2 Report

·         Line 11, “non-toxic and less energy intensive process” be more specific to the point in abstract.

·         Include more numerical importance in abstract.

·         Why Cr(VI) and Cd(II)? What wastewaters laden with these ions. Please elaborate. Why not Cr(III)?

·         What are the current treatments for Cr(VI) and Cd(II)? Their limitations that lead to your work.

·         Literature review should include more performances of adsorbents to remove Cr(VI) and Cd(II) from other studies.

·         Eq 2 can be removed. Simple formula.

·         The experiments involving the adsorption of Cr(VI) and Cd(II) are limited.

·         Characteristics of sludge used as precursor?

·         TGA results may not be useful unless you can link to the performances to adsorb Cr(VI) and Cd(II).

·         What are the industrial Cr(VI) and Cd(II) wastewater? And are those related to your studied concentrations?

·         Conclusions are too long and should be in statement forms of your important findings.

Author Response

The attached document contains the responses to the comments from the reviewer 2.

Round 2

Reviewer 2 Report

Well done on revision.